# Lung ultrasound is associated with distinct clinical phenotypes in COVID-19 ARDS: A retrospective observational study

Roy Rafael Dayan[1]*, Maayan Blau[1], Jonathan Taylor[2], Ariel Hasidim[1], Ori Galante[1,3], Yaniv Almog[1,3], Tomer Gat[1], Darya Shavialiova[1], Jacob David Miller[1], Georgi Khazanov[1], Fahmi Abu Ghalion[1], Iftach Sagy[1,4], Itamar Ben Shitrit[1,4], Lior Fuchs[1,3]

1 Faculty of Health Sciences, Ben-Gurion University of the Negev, Beersheba, Israel, 2 Intensive Care Unit, Soroka University Medical Center, Beersheba, Israel, 3 Division of Pulmonary, Critical Care, and Sleep Medicine, Mount Sinai Hospital, Icahn School of Medicine at Mount Sinai, New York, New York, United States of America, 4 Clinical Research Center, Soroka University Medical Center, Beersheba, Israel

☯ These authors contributed equally to this work.
* Roda2323@gmail.com

**Data Availability Statement:** All relevant data are within the paper and its Supporting Information files.

## Abstract

### Background

ARDS is a heterogeneous syndrome with distinct clinical phenotypes. Here we investigate whether the presence or absence of large pulmonary ultrasonographic consolidations can categorize COVID-19 ARDS patients requiring mechanical ventilation into distinct clinical phenotypes.

### Methods

This is a retrospective study performed in a tertiary-level intensive care unit in Israel between April and September 2020. Data collected included lung ultrasound (LUS) findings, respiratory parameters, and treatment interventions. The primary outcome was a composite of three ARDS interventions: prone positioning, high PEEP, or a high dose of inhaled nitric oxide.

### Results

A total of 128 LUS scans were conducted among 23 patients. The mean age was 65 and about two-thirds were males. 81 scans identified large consolidation and were classified as "C-type", and 47 scans showed multiple B-lines with no or small consolidation and were classified as "B-type". The presence of a "C-type" study had 2.5 times increased chance of receiving the composite primary outcome of advanced ARDS interventions despite similar SOFA scores, Pao2/FiO2 ratio, and markers of disease severity (OR = 2.49, %95CI 1.40–4.44).

### Conclusion

The presence of a "C-type" profile with LUS consolidation potentially represents a distinct COVID-19 ARDS subphenotype that is more likely to require aggressive ARDS

**Funding:** The author(s) received no specific funding for this work.

**Competing interests:** The above consulting relationship had no bearing or impact on the research conducted and the preparation of the manuscript. This does not alter our adherence to PLOS ONE policies on sharing data and materials. All other authors have no conflicts of interest to declare.

interventions. Further studies are required to validate this phenotype in a larger cohort and determine causality, diagnostic, and treatment responses.

## Introduction

The coronavirus disease 2019 (COVID-19) pandemic is a global crisis, that has challenged healthcare and economic systems worldwide [1]. COVID-19, caused by the severe acute respiratory syndrome coronavirus 2 (SARS-CoV-2) virus, predominantly causes viral pneumonia, with severity ranging from asymptomatic infection to fatal respiratory failure and acute respiratory distress syndrome (ARDS) [2]. The significant disease heterogeneity and potential for rapid deterioration mandates an objective, quick, bedside lung assessment tool that helps assess illness severity and guide clinical decisions.

Recent years have brought increasing recognition that ARDS represents a complex and heterogenous syndrome, with distinctive anatomic, physiologic, and molecular patterns [3]. Landmark work by Calfee and colleagues has demonstrated distinct ARDS phenotypes that respond differently to therapeutic interventions such as high PEEP and liberal fluid strategies [4,5]. Radiographic subtypes of ARDS, delineated as nonfocal/diffuse and focal/lobar, have also been described [6–10]. The distribution of pulmonary opacities and degree of lung consolidation represent variables with the potential to change lung mechanics and impact mechanical ventilation strategies [11]. In the LIVE trial, a personalized PEEP and prone positioning strategy based on radiographic sub-phenotypes achieved a reduction in 90-day mortality when only considering per-protocol treated patients [10].

In COVID-19 pneumonia, lung ultrasound (LUS) correlates with chest computerized tomography (CT) and is superior to chest X-ray in detecting pulmonary pathologies [12–15]. Chest CT findings have also been associated with COVID-19 pneumonia severity [16], collectively suggesting LUS may serve as a surrogate that is accurate, safe, and can be performed bedside. The characteristic LUS findings in COVID-19 patients involve B-lines, consolidations, and pleural irregularities (S1 Fig) [17]. These sonographic lung artifacts collectively have been associated with mortality and Intensive Care Unit (ICU) admission [18–20]; lung consolidations were specifically associated with critical illness [21,22]. A recent study by our group [23] described a simple clinical sonographic score based on known point-of-care ultrasound (POCUS) lung findings of COVID-19 pneumonia patients. The score, called "Point-of-care ultrasound Lung Injury Score" (PLIS) (S2 Fig and S1 Table) correlated with SOFA score, ICU admission, and mortality in patients with COVID-19 pneumonia, and was conducted successfully by novice operators to facilitate clinical patient care. In this follow-up study, we aim to investigate whether the sonographic detection of a large consolidation compared to either small or no consolidations, as represented by the C score component of the PLIS, can help stratify and phenotype patients with COVID-19 ARDS requiring mechanical ventilation. We examined whether patient PLIS C-score grades were associated with different outcomes, including respiratory parameters, mechanical ventilation characteristics, and response to therapeutic interventions.

## Methods

This observational retrospective cohort study was conducted at the Soroka University Medical Center, a tertiary care academic medical center in southern Israel. The data was collected during the COVID-19 pandemic among patients admitted to the medical intensive care unit

(ICU) between April 1st and September 30th, 2020. De-identified data was accessed between June 2021 and May 2022. Patients were included in the study if they had a diagnosis of COVID-19 confirmed via positive polymerase chain reaction for SARS-COV2 virus, required invasive mechanical ventilation, and were diagnosed with ARDS as set forth in the Berlin criteria [24]. Patients treated with extracorporeal membrane oxygenation in their clinical course were excluded. This study was approved by the institutional ethics committee (IRB #0195–20) with a waiver of informed consent. PLIS scans were completed by senior physicians experienced with LUS scans and internal medicine residents who received dedicated training with the LUS PLIS protocol. LUS scans were performed bedside during morning rounds as part of routine clinical assessment as an extension of the physical exam, as well as during specific clinical deterioration events, and findings were formally documented in the medical record. The ultrasound machines used were VENUE GO, GE Healthcare, R2 version. LUS scans were conducted with the 3SC phased-array probe, using the lung preset as set by the manufacturer to detect B-lines, and the cardiac preset to detect lung consolidations.

A total of 128 PLIS scans were conducted among 23 mechanically ventilated COVID-19 ARDS patients during the study duration. The PLIS scans were classified as "B-type" or "C-type" based on the value of the C component of the PLIS score. The "C-type" represents COVID-19 patients requiring mechanical ventilation with large LUS sonographic consolidations, defined as a consolidation measuring over 4 cm in the largest diameter (PLIS C-score classification of C2). All ultrasound images evaluating for consolidations were obtained in the mid-posterior axillary line above the diaphragm (Fig 1, designated zone 2). We classified the "B-type" phenotype as those who had minimal to small consolidations (less than 4 cm at greatest diameter) and met the criteria for a PLIS C-score of C0 or C1. L.R., an attending intensivist with over 10 years of LUS experience reviewed and agreed with the classifications.

Each patient could shift between the "B type" and "C type" phenotypes throughout their clinical course, according to the findings from the specific PLIS scan conducted that day. Real-time sonographic results and PLIS scores were not used to guide clinical practice but were collected for isolated research purposes to assess for clinical associations. Other variables collected included demographic characteristics, vital signs, laboratory results, admission characteristics, ventilation, and physiologic parameters. Oxygenation was assessed by measuring the PaO2/FiO2 ratio, while ventilation was estimated by using the Bohr equation, with Enghoff's

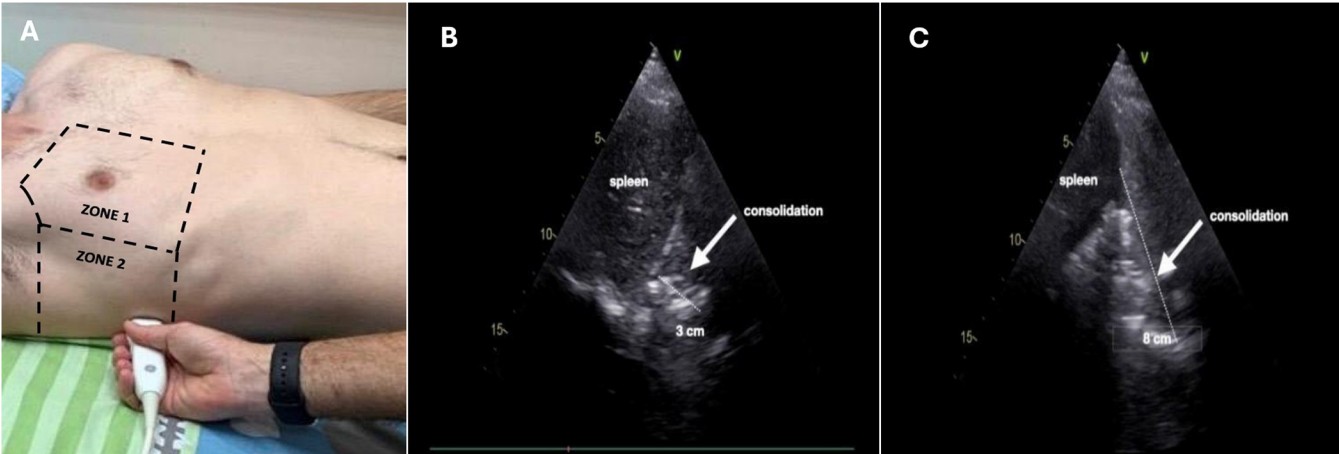

**Fig 1. Ultrasound image acquisition and sonographic sub-pleural lung consolidations.** The figure depicts the location of ultrasound image acquisition, situated at the mid-posterior axillary line above the diaphragm (A), alongside two distinct sonographic sub-pleural lung consolidations. These consolidations exhibit varying sizes: A small consolidation (B) and a larger consolidation (C).

modification [25]. Data was matched to each PLIS study according to the time the sonographic scan was conducted. Our primary outcome was a composition of three cardinal ARDS interventions: prone positioning, a high PEEP ($\geq$ 12 mmHg), and a high dose of inhaled nitric oxide (NO) ($\geq$ 15 ppm). This composite was selected to represent interventions associated with more significant ARDS illness ("advanced ARDS interventions"). Secondary outcomes included PaO2/FiO2 ratio and other respiratory parameters.

Data is expressed as mean ± standard deviation (SD), median ± interquartile range (IQR), or number and percentage. Each single PLIS scan obtained from the patients who met inclusion criteria composed the unit used for analysis. Patients were stratified by "B-type" versus "C-type" PLIS, and data was analyzed using the t-test for continuous variables, the chi-square for dichotomous variables, and parametric tests for ordinal variables. Multivariate generalized estimating equation (GEE) regression was used to evaluate the covariates associated with the study outcomes: PEEP, inhaled NO doses, and prone position—a composite outcome of the above. The GEE model allows the analysis of repeated measures with non-normal response variables, including the primary outcome. The final model was selected based on the plausible clinical explanation, statistical significance, and goodness of fit. The statistical analysis was conducted using SPSS version 25.0.

## Results

A total of 47 PLIS scans were classified as "B-type" studies (owing to a C-score of C0-C1), and 81 scans were classified as "C-type" studies (C-score of C2). Patients' baseline characteristics and admission information are presented in Table 1. The average age of patients in the cohort was 65.3 years (±11.9) and about two-thirds were males (69.9%, n = 16). Mean body mass index (BMI) was 27.3 (±3.9), 47.8% (n = 11) had diabetes mellitus, and 17.4% (n = 4) from chronic obstructive pulmonary disease. The median ICU length of stay was 17 days and the mortality in the cohort was 61%.

Table 2 compares the clinical characteristics, ventilatory parameters, and therapies at the time of POCUS scanning and PLIS acquisition, comparing "B-type" and "C-type" studies (for complete and comprehensive cohort outcomes, see S2 Table).

**Table 1. General characteristics of the cohort.**

| Variable | All Patients (n = 23) |
|---|---|
| Age (years) (mean ± SD) | 65.3 ± 11.9 |
| Males n(%) | 16 (69.9%) |
| Smoking n(%) | 5 (21.7%) |
| Body Mass Index (mean ± SD) | 27.3 ± 3.9 |
| Diabetes n(%) | 11 (47.8%) |
| Chronic obstructive pulmonary disease n(%) | 4 (17.4%) |
| Malignancy n(%) | 3 (13%) |
| Chronic kidney disease n(%) | 2 (8.7%) |
| Cerebrovascular disease n(%) | 2 (8.7%) |
| Congestive heart failure n(%) | 0 (0) |
| Mortality n(%) | 14 (60.9%) |
| Hospitalization days in Intensive Care Unit (median, interquartile range) | 17 (7–32) |
| Total hospitalization days (median, interquartile range) | 24 (14–38) |
| Days on mechanical ventilation (median, interquartile range) | 16 (6–27) |

**Table 2. Clinical characteristics, respiratory parameters, and interventions regarding B-type versus C-type patterns during the PLIS scan.**

| Variable | B-Type scans (n = 47) | C-Type scans (n = 81) | P-value |
|---|---|---|---|
| **Clinical Characteristics and Laboratory Results** | | | |
| Heart rate (mean ± SD) (bpm) | 94 ± 15 | 92 ± 11 | 0.5 |
| Mean arterial pressure (mean ± SD) (mmHg) | 83 ± 37 | 76 ± 39 | 0.36 |
| Systolic blood pressure (mean ± SD) (mmHg) | 136 ± 21 | 128 ± 18 | 0.05 |
| Diastolic blood pressure (mean ± SD) (mmHg) | 78 ± 11 | 77 ± 14 | 0.63 |
| Temperature (mean ± SD) (Celsius) | 37.2 ± 0.8 | 37.2 ± 0.6 | 0.61 |
| Oxygen saturation % (mean ± SD) | 91 ± 7 | 91 ± 3 | 0.62 |
| International Normalized Ratio (mean ± SD) | 1.28 ± 0.3 | 1.15 ± 0.14 | 0.12 |
| Serum Albumin (mean ± SD) (g/dl) | 2.53 ± 0.38 | 2.23 ± 0.38 | **0.015** |
| Total Bilirubin (mean ± SD) (mg/dl) | 0.48 ± 0.23 | 0.48 ± 0.3 | >0.9 |
| Platelet count (mean ± SD) (per µL of blood) | 325 ± 110 | 348 ± 124 | 0.3 |
| Serum Creatinine (mean ± SD) (mg/dl) | 1.11 ± 1.13 | 0.85 ± 0.66 | 0.11 |
| SOFA score (median, interquartile range) | 5 (4–7) | 6 (4–7) | 0.39 |
| Total PLIS (median, interquartile range) | 2 (1–3) | 4 (3–4) | **<0.001** |
| Lactate (median, interquartile range) (mmol/L) | 1.4 (1.2–2.1) | 1.9 (1.2–2.3) | 0.14 |
| **Ventilation and Respiratory Parameters** | | | |
| PaO2/FiO$_2$ (median, interquartile range) | 178 (123–242) | 137 (95–193) | 0.06 |
| PaCO$_2$ (median, interquartile range) (mmHg) | 50 (44–58) | 52 (45–57) | 0.29 |
| End-Tidal CO$_2$ (median, interquartile range) (mmHg) | 32 (29–36) | 35 (31–38) | **0.01** |
| Respiratory rate (median, interquartile range) (breaths per minute) | 20 (15–24) | 20 (16–24) | 0.26 |
| Tidal volume (median, interquartile range) (ml) | 555 (500–600) | 535 (500–580) | 0.28 |
| Minute ventilation (median, interquartile range) (ml/minute) | 10,000 (7,650–12,440) | 10,700 (8,580–12,000) | 0.8 |
| Dead space (vd/vt) (% from Tidal volume) | 33 (23–40) | 32 (26–42) | 0.86 |
| Peak pressure (median, interquartile range) | 17 (15–20) | 19 (15–23) | 0.42 |
| **ARDS Interventions** | | | |
| Prone position n(%) | 3 (6.4) | 14 (17.3%) | 0.08 |
| PEEP (median, interquartile range) (C$_m$H$_2$O) | 10 (6–13) | 12 (10–14) | 0.14 |
| NO administration n(%) | 9 (19.1) | 29 (35.8%) | **0.047** |
| NO dose (median, interquartile range) (ppm) | 15 (9–16) | 15 (13–17) | 0.59 |
| Advanced ARDS interventions* n(%) | 19 (40.4) | 48 (59.3%) | **0.04** |
| Vasopressor administration n(%) | 36 (76.6) | 70 (86.4%) | 0.22 |

ARDS- Acute Respiratory Distress Syndrome; FiO2—Fractional-inspired oxygen; NO—nitric oxide; PLIS—Point-of-care Lung Ultrasound Injury Score; PaCO2- Partial pressure of CO2 in arterial blood; PaO2- Partial pressure of oxygen in arterial blood; PEEP—positive end-expiratory pressure; SOFA score—Sequential Organ Failure Assessment score.

*Advanced ARDS interventions–a composite outcome of prone positioning, high PEEP ($\geq$ 12 C$_m$H$_2$O), or high dose of inhaled NO ($\geq$15 ppm).

The median PLIS was higher in the "C-type" group as anticipated (4 vs 2, p <0.001) while the median SOFA score did not differ significantly between groups (SOFA score of 6 versus 5 points, p = 0.39). The median PaO2/FiO2 ratio was 178 in the "B-type" as compared to 137 in the "C-type" group, a difference that was not statistically significant (p = 0.06). Ventilation parameters were similar between the two sonographic groups; peak inspiratory pressures (PIP) and tidal volumes (TV) were unchanged, with a median PIP of 19 C$_m$H$_2$O vs 17 C$_m$H$_2$O, (p = 0.42) and TV of 555 ml vs 535 ml (p = 0.28). The primary composite outcome of applied high levels of PEEP ($\geq$ 12 C$_m$H$_2$O), high doses of inhaled NO ($\geq$15 ppm) or proning was found at significantly higher rates in patients with a "C-type" study (59% vs 40%, p = 0.04). Analyzing the components of the composite outcome separately, patients with the "C-type"

**Table 3. Multivariate Generalized Estimating Equation (GEE) regression for the composite outcome of advanced ARDS interventions\*.**

| Variable | OR | P-value | 95% Confidence interval |
|---|---|---|---|
| Age | 0.99 | 0.82 | 0.91–1.07 |
| Body mass index | 1.04 | 0.81 | 0.72–1.51 |
| PaO2/FiO2 | 1.01 | **0.01** | 1.00–1.02 |
| Diabetes | 0.24 | 0.13 | 0.04–1.15 |
| C-type | 2.49 | **0.01** | 1.40–4.44 |

FiO2- Fractional inspired oxygen; PaO2- Partial pressure of oxygen in arterial blood.

\*Advanced ARDS interventions–a composite outcome of prone positioning, high PEEP ($\geq$ 12 $C_mH_2O$), or high dose of inhaled NO ($\geq$15 ppm).

phenotype were significantly more likely to be treated with inhaled NO at the time of sonography acquisition (36% vs 19%, p = 0.047), and were more likely to be placed in the prone position, though this specific difference in positioning did not meet the threshold for significance (17% vs 6%, p = 0.08). Applied PEEP alone at the time of PLIS scanning did not differ significantly between groups.

Table 3 describes a generalized multivariate estimation (GEE) model for the odds of being treated with the advanced ARDS therapeutic interventions including prone position, high PEEP, or high doses of inhaled NO therapy for "C-type" versus "B-type" phenotypes. Comorbid diabetes mellitus, age, BMI, and PaO2/FiO2 ratio did not significantly increase the risk for the need for these interventions. However, possessing a "C-type" PLIS sonographic study pattern was found to significantly increase the odds of receiving such an intervention as represented by the composite outcome by 2.5 times compared to having a "B-type" study (OR = 2.49, CI = 1.40–4.44).

Fig 2 depicts the course of ten selected COVID-19 ARDS patients on mechanical ventilation, and visually illustrates their clinical status, ARDS interventions required, and the corresponding LUS sub-group ("C-type" vs "B-Type" studies) during their ICU stay. Patients with the most available PLIS data were selected to represent the study concept visually.

## Discussion

In this follow-up study to our original description of the PLIS, we aim to further categorize heterogenous COVID-19 ARDS patients requiring mechanical ventilation and identify distinctive sonographic phenotypes according to the presence ("C-type") or absence ("B-type") of large lung consolidation by bedside ultrasound. Our previous work [23] concerning hospitalized COVID-19 pneumonia patients described the PLIS. This new simple clinical-sonographic lung ultrasound score is quick, reproducible, and easy to perform at the bedside, and has been shown to correlate with the SOFA score, ICU rate of admission, and the risk of in-hospital mortality. Here we further categorized patients into two broad sonographic lung patterns based on the burden of lung consolidation. Our results demonstrate that for similar SOFA scores, PaO2/FiO2 ratios, and markers of clinical illness severity, those characterized by a "C-type" PLIS study were more likely to be treated with inhaled NO at the time of POCUS scanning and had a significantly increased likelihood of concurrently receiving a composite outcome of high PEEP, high dose of inhaled NO (>15 ppm) or ventilation in the prone position (OR = 2.49). Our data suggests that the "C-type" profile of COVID-19 ARDS patients represents a distinct sub-phenotype of COVID-19 ARDS, irrespective of illness severity, that may be more likely to require advanced therapies to improve oxygenation.

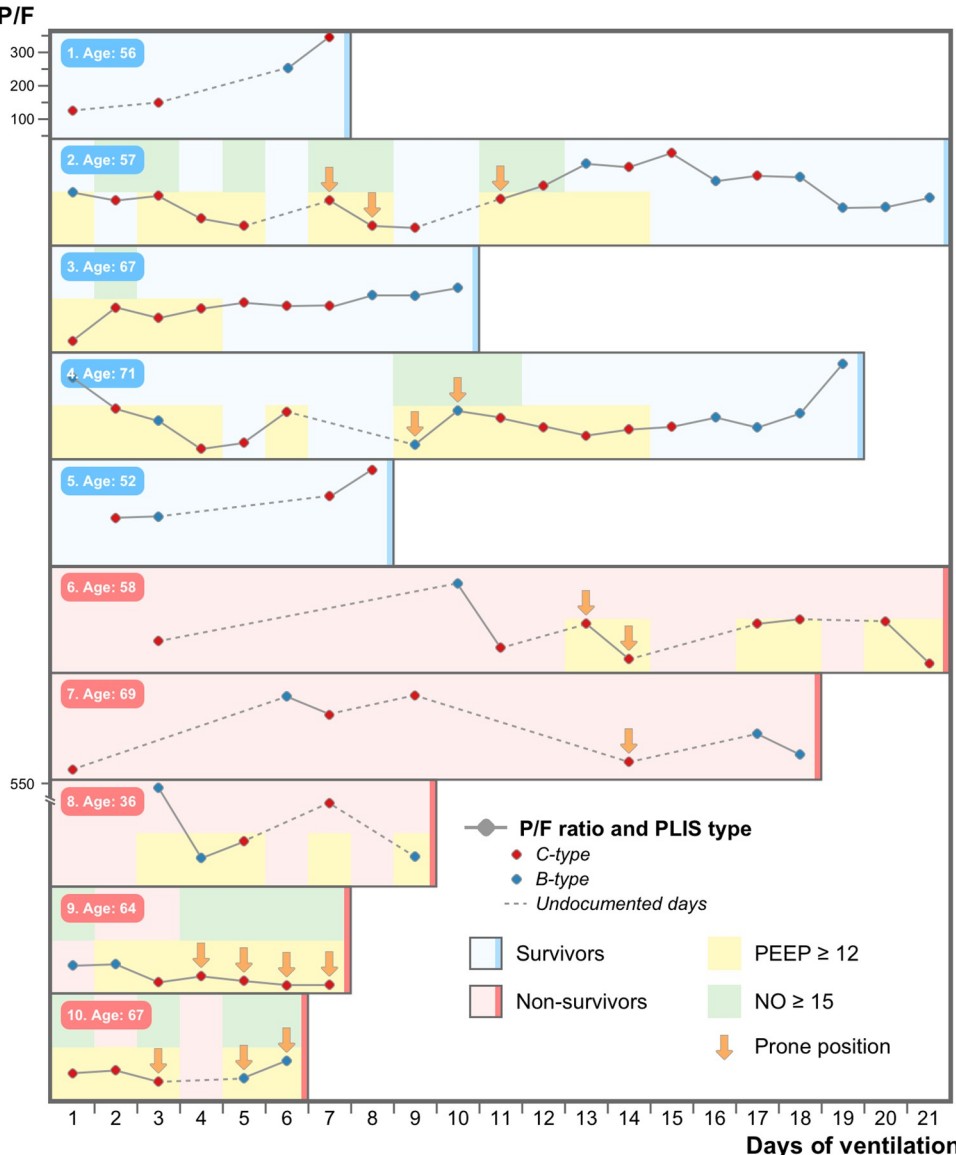

**Fig 2. The course of ten representative COVID-19 ARDS ventilated patients\* during their Intensive care unit hospitalization.** The figure graphically represents the course of ten ventilated patients with COVID-19 ARDS. half survived (in blue) and half died (in red). The PaO2 to FiO2 ratio is plotted on the y-axis and the days of mechanical ventilation on the x-axis. Each circle represents a single PLIS study performed (red circle for the "c-type" and blue circle for the "b-type"). Vertical arrows signify patients ventilated with prone positioning. Backgrounds colored yellow and green represent patients currently ventilated with a high PEEP (12 cmH2o or more) and a high dose of inhaled nitric oxide (15ppm or above), respectively. \*Patients with the most available PLIS data were selected to visually represent the cohort.

Fig 2 graphically delineates the course of ten representative patients during their ICU stay to illustrate the study concept. Those with the most PLISs available were chosen for the purpose of visual representation. As a representative example, patients #2, #,3, and #4 (all survivors) possessed mostly the "C-Type" profiling at the beginning of their critical illness, which was characterized by severe hypoxemia and more advanced ARDS interventions, relative to the end of their clinical course, which was characterized with "B-type" phenotype and fewer interventions. The opposite was true regarding patients #6 and #9 (non-survivors): near the

end of their course, they were characterized mostly by "C-type" lung injury and required advanced interventions.

As the field of critical care increasingly moves towards precision medicine, several authors have sought to profile heterogeneous lung injury in COVID-19 ARDS. The Gattinoni model [26] early in the pandemic proposed two different phenotypes on a time-related disease spectrum: The "L-phenotype" and the "H-phenotype". The latter is characterized by high lung weight, and high consolidation burden, with a relatively high potential for recruitment of non-aerated lung areas. Although this has subsequently been challenged [27], conceptually, the "H-phenotype" may parallel the "C-type" of the PLIS and could potentially be identified early by LUS. In a landmark study in ARDS before the COVID-19 pandemic, latent class analysis identified two distinct ARDS phenotypes: one termed "hyperinflammatory", associated with more inflammatory cytokines, lower serum bicarbonate, and higher vasopressor requirements, and a different "hypoinflammatory" phenotype, with lower concentrations of cytokines, higher bicarbonate, and decreased vasopressor needs [4]. These phenotypes have been validated in multiple large observational cohorts, with post hoc analysis of landmark ARDS trials demonstrating differing responses to randomized PEEP strategies between phenotypic profiles [4]. This categorization has extended to patients with COVID-19, where the hyperinflammatory phenotype showed an improved response to corticosteroids [28]. Radiographic subphenotypes of ARDS have similarly been described, stratified into nonfocal and lobar types [5–7]. Nonfocal/diffuse ARDS has been associated with worse lung compliance, higher mortality, and lower levels of sRAGE, a plasma biomarker of epithelial damage [8]. Similarly, in a per-protocol analysis of the LIVE trial, a personalized PEEP and prone positioning strategy based on radiographic sub-phenotypes achieved a reduction in 90-day mortality [10]. These concepts have been extended to COVID-19 ARDS, as latent class analysis has revealed multiple COVID-19 ARDS sub-phenotypes, stratified by dead space ventilation and mechanical power, with important patient-related downstream outcomes [29]. Together, this work demonstrates that subgroups of ARDS may have differential responses to therapy and underscores the importance and urgency of research to prospectively identify treatment-responsive subgroups, along with markers to identify them at the bedside.

Lung ultrasound is ripe for sonographic ARDS phenotyping. Wang et al. [30] showed that in non-COVID-19 ARDS patients, response to prone positioning can be predicted by the LUS score composed of 16 scanning sites. This has subsequently been replicated in awake and spontaneously breathing severe COVID-19 patients [31]. Bouhemad et al. [32] demonstrated a significant correlation between PEEP-induced lung recruitment as measured by pressure-volume curves and an ultrasound reaeration score. Lichter et al. [33] showed an association between worsening LUS scores and increased PEEP requirements in COVID-19 ARDS ventilated patients, though the score used in this study consisted of a complicated 12-point scanning protocol.

The totality of the evidence suggests that lung ultrasound has an important role to play in further characterizing and phenotyping this population, and perhaps guiding interventions. The strength of the PLIS protocol lies in its feasibility and simplicity. While the above studies used more complicated and time-consuming LUS scores based on multiple scanning points, we dichotomized the lung injury status based solely on the presence of sonographic sub-pleural consolidations. A "C-type" scanning profile, consisting of a LUS consolidation measuring over 4 cm in the largest diameter, was associated with higher rates of concurrently receiving a composite of high PEEP, high doses of inhaled NO, or prone positioning, regardless of illness severity. We believe this finding is thought-provoking and should be further studied on a larger scale with a controlled population. A proposed update to the global definition of ARDS included the use of ultrasound as an acceptable imaging modality [34], and the new 2023

European Society of Intensive Care Medicine guidelines on ARDS [35] explicitly call for simple, real-time, and rapid tests to aid in ARDS subphenotype classification in prospective studies. We posit, and feel that our results demonstrate, that this is a role that lung ultrasound is well suited for.

Our study has several limitations. It is a small and observational retrospective study, performed in a single center large academic medical ICU. Although a significant difference in the primary outcome between the two study groups was demonstrated, some of the separate individual components of the composite failed to reach statistical significance, likely due to the small sample size. While many different clinical factors go into the decision to prone patients, initiate pulmonary vasodilators, or use a high PEEP strategy, these clinical interventions comprising the composite primary outcome are all associated with a more severe ARDS illness course. Thus, the significant association with large sonographic subpleural consolidations still serves as an important finding, despite the limitations of using a composite measurement. Additionally, the single-center nature of our cohort limits generalizability and external validity. Our study was also not blinded, as the physicians who conducted the PLIS scans were the same treating physicians making decisions regarding patient management. However, the presence and size of consolidations during the PLIS scan were not considered in clinical decision-making due to the lack of clarity regarding how to appropriately translate LUS findings into practical clinical care, and thus the PLIS was conducted and recorded prospectively solely for research purposes. The observational nature of these findings does not allow any conclusions regarding causality and only suggests associations, while the ultimate goal would be to predict which subset of patients may respond to specific therapies or recruitment maneuvers. To this end, we recommend a prospective clinical investigation to further assess if the "C-type" phenotype has a relatively more favorable response to therapeutic interventions such as high PEEP, inhaled NO, or prone positioning.

## Conclusions

The PLIS protocol is a simple and reproducible LUS tool that can characterize and phenotype COVID-19 ARDS patients requiring invasive mechanical ventilation based on the presence or absence of large sonographic sub-pleural lung consolidations, independent of clinical disease severity. Patients characterized with the "C-type" scanning profile, with large consolidations on ultrasound, were more likely to require advanced ARDS interventions as represented by a composite outcome of high PEEP, high dose of inhaled NO, or prone positioning. Further larger-scale studies are required to assess the causality of this relationship, its association with physiologic parameters like mechanical power and dead space, and its potential for targeted therapeutic interventions.

## Supporting information

**S1 Fig. The classic ultrasonographic lung findings in COVID-19 patients.**
(TIF)

**S2 Fig. Areas of lung ultrasound scanning.**
(TIF)

**S1 Table. The Point of Care Lung Ultrasound Injury Score (PLIS) grading system.**
(DOCX)

**S2 Table. A comprehensive cohort data for each patient.**
(DOCX)

## Author Contributions

**Conceptualization:** Roy Rafael Dayan, Ori Galante, Yaniv Almog, Lior Fuchs.

**Data curation:** Maayan Blau, Tomer Gat, Darya Shavialiova, Jacob David Miller, Georgi Khazanov, Fahmi Abu Ghalion, Iftach Sagy.

**Formal analysis:** Ariel Hasidim.

**Methodology:** Iftach Sagy, Lior Fuchs.

**Supervision:** Lior Fuchs.

**Validation:** Lior Fuchs.

**Writing – original draft:** Roy Rafael Dayan, Maayan Blau, Jonathan Taylor.

**Writing – review & editing:** Roy Rafael Dayan, Maayan Blau, Jonathan Taylor, Itamar Ben Shitrit, Lior Fuchs.

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
