## [Decision Letter · Decision Letter 0]

13 Nov 2023

PONE-D-23-24480Lung ultrasound is associated with distinct clinical phenotypes in COVID-19 ARDS: A retrospective observational studyPLOS ONE

Dear Dr. Dayan,

Thank you for submitting your manuscript to PLOS ONE. After careful consideration, we feel that it has merit but does not fully meet PLOS ONE’s publication criteria as it currently stands. Therefore, we invite you to submit a revised version of the manuscript that addresses the points raised during the review process.

Three experts in the subject matter reviewed your paper. Although the reviewers noted the importance of the topic, they expressed major concerns about the manuscript.

The authors performed a retrospective study involving 23 patients with COVID-19-related ARDS, proposing lung ultrasound findings as an imaging marker for ARDS phenotyping. The manuscript is well-written and organized. The reviewers emphasized the authors' work of performing multiple scans and analyzing the images and the role lung imaging can play in ARDS phenotyping.

A common concern among reviewers was the lack of details about the criteria/methods for patient selection, imaging scores, and scans included in the analysis since multiple measurements were performed for each enrolled patient. In addition, there is a need for more clarification about the statistical approach used for the repeated measurement. Finally, I'd like to point out that an essential observation addressed is the limitation of a composite outcome formed by a combination of interventions, and it should be addressed more in-depth throughout the manuscript.

I look forward to your revision.Please ensure that your decision is justified on PLOS ONE’s publication criteria and not, for example, on novelty or perceived impact.

We look forward to receiving your revised manuscript.

Kind regards,

Roberta Ribeiro De Santis Santiago, M.D., Ph.D., R.R.T.

Academic Editor

PLOS ONE

“I have read the journal's policy and the authors of this manuscript have the following competing interests: Lior Fuchs is a consultant of General Electric Healthcare. All other authors have no conflicts of interest to declare.”

We note that one or more of the authors are employed by a commercial company: General Electric Healthcare

Reviewers' comments:

Reviewer's Responses to Questions

**Comments to the Author**

1. Is the manuscript technically sound, and do the data support the conclusions?

Reviewer #1: Yes

Reviewer #2: No

Reviewer #3: Partly

2. Has the statistical analysis been performed appropriately and rigorously? 

Reviewer #1: Yes

Reviewer #2: Yes

Reviewer #3: I Don't Know

3. Have the authors made all data underlying the findings in their manuscript fully available?

Reviewer #1: Yes

Reviewer #2: Yes

Reviewer #3: No

4. Is the manuscript presented in an intelligible fashion and written in standard English?

Reviewer #1: Yes

Reviewer #2: Yes

Reviewer #3: Yes

5. Review Comments to the Author

Reviewer #1: Roy Rafael Dayan, M.D., and colleagues investigate the utility of lung ultrasound (using a previously published scoring system) in distinguishing clinical phenotypes among patients with ARDS caused by COVID-19 and its associations with clinical outcomes.

The text is well written, however I have some mayor andminor recommendations

Mayor reviews:

# Multiple evaluations of LUS

-It is not clear from which point of the patient´s history the authors defined the main score of each patient. So, the final classification of each patient came from the first evaluation, maximum score, etc?

- How was the multivariable model fit? Using repeated measures? Please, clarify.

- How can a clinician know which profile a patient has? For instance, a patient score B, B, and C? or B-C-C?

Minor reviews:

-This reviewer believes that the introduction is too large. It will be better to condense the introduction to provide a clear and concise overview of your study objectives. You will expand upon the detailed information in the discussion section.

Reviewer #2: Major Comments

-Overall, this study is well-written and organized. The Discussion in particular provides a thorough summary of existing literature and ties in the study findings nicely. I also commend the authors on performing the arduous task of performing and interpreting serial lung ultrasound examinations. However, there are major flaws with the study design that would need to be addressed.

-The author’s primary outcome is heavily flawed. The stated objective of this study was to investigate whether the presence of large consolidation on lung ultrasound alone was predictive of outcomes among COVID-19 ARDS patients. However, the primary outcome was not a measure of patient outcomes (ie. duration of mechanical ventilation, duration of ICU stay, mortality) but instead a composite of whether the patient required proning, high PEEP, or inhaled pulmonary vasodilator. The decision to employ one of these interventions is heavily biased by the treating physician and is often related to the patient’s underlying comorbidities rather than simply the severity or type of ARDS. For example, an obese patient would necessitate high PEEP regardless of the presence or severity of ARDS to facilitate alveolar recruitment and minimize atelectasis, while a patient with pulmonary hypertension may require an inhaled pulmonary vasodilator even with very mild ARDS. The combination of these 3 very different interventions into a composite outcome further weakens the study findings, as stated in the Discussion (Lines 281 – 284).

-Phenotyping in ARDS has typically referred to the underlying molecular mechanisms involved in the lung damage, based on local and systemic biomarkers, and degree of inflammation. Here, the authors instead use the term phenotype to describe whether or not the patient had large consolidations on lung ultrasound. The use of “phenotyping” in this manuscript is misleading and should be avoided or specified as “sonographic phenotyping”, as the presence of consolidation does not constitute a molecular phenotype but simply describes an imaging finding.

- In the methods, the authors state (line 139-140): " A single PLIS scan per patient composed the unit used for analysis”. This seems to be incorrect, as 128 data points (47 B-type and 81 C-type) scans were included in the results. Please clarify. If a single scan was used for each patient, please describe which scan was selected for analysis and how this selection was justified.

-The patients performed serial ultrasound examinations on each patient, which provides a robust data set when looking at each individual patient longitudinally (as shown in Figure 1) but allows for confounding if treating each scan as a separate data point, as they appear to be.

-Figure 1 is very well designed and formatted to capture the clinical courses of these patients. However, the conclusion that is drawn from this data seems contrary to the author’s main conclusion, showing that the non-survivors tended to develop consolidations while the survivors tended to have isolated B-lines without consolidation. This would imply that consolidations are associated with more severe disease, which could very well be true and this study was simply underpowered and not properly designed to detect this.

Minor Comments

-Lines 23, 32: please avoid or define non-standard abbreviations in the abstract, including “PLIS” and “LUS”

-Lines 31, 130: please use appropriate capitalization and subscript for “PaO2/FiO2 ratio”

-Line 42: the authors should rephrase “asymptomatic carriage”; could simply say “asymptomatic”

-Lines 44-45: I would argue against the description of COVID-19 as “ambiguous clinical presentation”. Rather, these patients routinely present with symptoms of dyspnea and increased work of breathing. Instead, I would describe how these patients often present similarly but early in their course it can be difficult to differentiate those who will progress to severe disease.

-Line 53: use distinct rather than distinctive here

-Line 54: I believe you mean restrictive fluid strategies (rather than liberal), as liberal use of fluids is not a therapeutic intervention in ARDS

-Line 66: insert comma after COVID-19. Also, this sentence needs to be shorted.

-Line 71: This should be a new sentence (after reference 18).

-Line 78: intensive care unit should not be capitalized

-Table 1: please reformat as the current formatting of data is confusing. Continuous data (mean ± SD) should be written as such. For example, age (years) should be written as 65.3 ± 11.9 rather than 65.3 (11.9). Non-continuous data should be written, for example for males, as 16 (69.9%) and specified as n (%) rather than n, %.

-Line 157: POCUS is used as an abbreviation and has not yet been defined.

Reviewer #3: It was a pleasure for me to read your manuscript and your previous work on the PLIS scanning protocol. I am personally a big fan of this thought process, and in general, I am an advocate for bedside phenotyping of ARDS. I suggest the methods to be explained a little bit more in detail in terms of patient selection. I sense that a certain amount of patients admitted to your center were screened, and that only patients that had "enough" scans were considered for the composite outcome. In my mind, it should be explained more clearly who was screened, and hence what were the inclusion and the exclusion criteria (I see ECMO clearly stated). Table 2 should have driving pressure, and tidal volume per kg of IBW among the variables. I like the concept of Figure 3, but if this is a study on 23 patients, I think they should all be presented instead of 10, either as supplemental material or in the main manuscript if physically possible. The choice for advanced ARDS interventions mentioned in the composite outcome seem to correlate with the C-phenotype more than oxygenation. But it is oxygenation what normally clinically drives these interventions. It makes me wonder whether in your center you include ultrasound evaluation to back up those decisions at rounds. I guess it could make sense in the setting of widespread ultrasound use and proficiency. Also, with this being a retrospective study, I imagine that there is a chance that the arterial blood gases do not always match the scans. These are all aspects that should be covered in the methods, in the discussion and in the conclusion. Finally, I would add a supplementary table about the advanced interventions with the PaO2/FiO2 ratio at the moment when the advanced intervention were started, what was the PLIS score that day, the group B vs C the patients "belonged to" that day, the average daily score from admission to advanced intervention, and the average daily score from intubation to advanced intervention, if data is available.

6. PLOS authors have the option to publish the peer review history of their article (what does this mean?). If published, this will include your full peer review and any attached files.

Reviewer #1: No

Reviewer #2: No

Reviewer #3: **Yes: **Raffaele Di Fenza

---

## [Author Response · Author response to Decision Letter 0]

24 Dec 2023

Dear PLOS One Editorial and Reviewing Team,

Dear PLOS One Editorial and Reviewing Team,

We are writing to express our sincere gratitude for the time and effort you invested in reviewing our manuscript. We value your insightful comments and suggestions, which have significantly contributed to improving the quality of our work.

We have carefully considered all of your feedback and have incorporated them into the revised manuscript. We have addressed each comment point-by-point in a separate document, which we have attached for your reference. We believe that the revised manuscript is now more complete.

We are pleased to present the revised manuscript to you for your further consideration. We are confident that it addresses all of your concerns and meets the high standards of PLOS One.

We appreciate your expertise and dedication to publishing high-quality research. We look forward to your feedback on the revised manuscript.

Thank you again for your time and consideration.

Sincerely,

Roy Rafael Dayan

---

## [Decision Letter · Decision Letter 1]

13 Feb 2024

PONE-D-23-24480R1Lung ultrasound is associated with distinct clinical phenotypes in COVID-19 ARDS: A retrospective observational studyPLOS ONE

Dear Dr. Dayan,

Thank you for submitting your manuscript to PLOS ONE. After careful consideration, we feel that it has merit but does not fully meet PLOS ONE’s publication criteria as it currently stands. Therefore, we invite you to submit a revised version of the manuscript that addresses the points raised during the review process.

Although the reviewers acknowledged the improvement of the manuscript, there are still some issues. In particular, the manuscript should report more details about the ultrasound evaluation. Also, a more concise introduction would improve the manuscript.

We look forward to receiving your revised manuscript.

Kind regards,

Francesca Pennati, Ph.D.

Academic Editor

PLOS ONE

Journal Requirements:

Additional Editor Comments:

According to the second reviewer, more details about the ultrasound evaluation should be provided. Also, a more concise introduction would improve the manuscript.

Reviewers' comments:

Reviewer's Responses to Questions

**Comments to the Author**

1. If the authors have adequately addressed your comments raised in a previous round of review and you feel that this manuscript is now acceptable for publication, you may indicate that here to bypass the “Comments to the Author” section, enter your conflict of interest statement in the “Confidential to Editor” section, and submit your "Accept" recommendation.

Reviewer #1: (No Response)

Reviewer #3: All comments have been addressed

2. Is the manuscript technically sound, and do the data support the conclusions?

Reviewer #1: Partly

Reviewer #3: Yes

3. Has the statistical analysis been performed appropriately and rigorously? 

Reviewer #1: Yes

Reviewer #3: I Don't Know

4. Have the authors made all data underlying the findings in their manuscript fully available?

Reviewer #1: Yes

Reviewer #3: Yes

5. Is the manuscript presented in an intelligible fashion and written in standard English?

Reviewer #1: Yes

Reviewer #3: (No Response)

6. Review Comments to the Author

Reviewer #1: Roy Rafael Dayan et al. submitted a paper presenting retrospective data on the efficacy of lung ultrasound in identifying distinct clinical phenotypes in COVID-19 ARDS, each with unique outcomes.

Major comments:

* The authors propose that patients may exhibit various phenotypes throughout their clinical course. However, the analysis of outcomes associated with phenotypes C and B appears to represent two distinct populations. It remains unclear at which point in the patients' clinical history the lung ultrasound (LUS) was conducted and when a patient was classified as having phenotype C or B.

* Although the group refers to a previous ultrasound score, providing a more detailed explanation of the ultrasound evaluation would enhance clarity. Specifically, it would be helpful to elaborate on which part of the thorax was assessed. For instance, determining whether the dorsal region, where larger consolidations are typically found, can be reliably assessed by ultrasound, especially in patients in the prone position, would be valuable. This aspect is crucial to the methodology and limitations, as an inability to evaluate a certain thoracic region via ultrasound may lead to a potential underestimation of the phenotype.

Minor Comments:

* The introduction is lengthy; it would benefit from being more concise. To achieve this, consider avoiding the repetition of studies and concepts already discussed in the subsequent sections.

Reviewer #3: Thank you for resubmitting. Thank you for clarifying the inclusion/exclusion criteria in your dataset. I agree that oxygenation should be only one small part of what triggers interventions, which is why I was hoping for more data regarding protective ventilation. But I appreciate your clarifications about the method and how hypothesis were generated.

US can complement CXR to quickly photograph patients that have less homogeneously ventilated lungs, shunt more, and trigger specific treatments. Please, keep up: an ultrasound-trained generation of intensivists is expected to multiply the number of patients and observations, correlate with other imaging techniques and patient's respiratory mechanics.

7. PLOS authors have the option to publish the peer review history of their article (what does this mean?). If published, this will include your full peer review and any attached files.

Reviewer #1: No

Reviewer #3: **Yes: **Raffaele Di Fenza

---

## [Author Response · Author response to Decision Letter 1]

16 Mar 2024

Please see the attached "Response to Reviewers" for a point-by-point response to the reviewers’ comments and concerns. 

Sincerely,

Dr. Roy Rafael Dayan

---

## [Editor Report · Decision Letter 2]

14 May 2024

Lung ultrasound is associated with distinct clinical phenotypes in COVID-19 ARDS: A retrospective observational study

PONE-D-23-24480R2

Dear Dr. Dayan,

We’re pleased to inform you that your manuscript has been judged scientifically suitable for publication and will be formally accepted for publication once it meets all outstanding technical requirements.

Kind regards,

Francesca Pennati, Ph.D.

Academic Editor

PLOS ONE
---

## [Editor Report · Acceptance letter]

25 May 2024

PONE-D-23-24480R2 

PLOS ONE

Dear Dr. Dayan, 

I'm pleased to inform you that your manuscript has been deemed suitable for publication in PLOS ONE. Congratulations! Your manuscript is now being handed over to our production team.

Kind regards, 

on behalf of

Dr. Francesca Pennati 

Academic Editor

PLOS ONE